# Variation in Hydrogel Formation and Network Structure for Telo-, Atelo- and Methacrylated Collagens

**DOI:** 10.3390/polym14091775

**Published:** 2022-04-27

**Authors:** Malachy Kevin Maher, Jacinta F. White, Veronica Glattauer, Zhilian Yue, Timothy C. Hughes, John A. M. Ramshaw, Gordon G. Wallace

**Affiliations:** 1Intelligent Polymer Research Institute, ARC Centre of Excellence for Electromaterials Science, AIIM Facility, Innovation Campus, University of Wollongong, Wollongong, NSW 2519, Australia; mkm290@uowmail.edu.au (M.K.M.); zyue@uow.edu.au (Z.Y.); 2CSIRO Manufacturing, Clayton, Melbourne, VIC 3168, Australia; jacinta.white@csiro.au (J.F.W.); veronica.glattauer@csiro.au (V.G.); timothy.hughes@csiro.au (T.C.H.); 3Department of Surgery, St. Vincent’s Hospital, University of Melbourne, Melbourne, VIC 3065, Australia; johnamramshaw@gmail.com

**Keywords:** collagen, hydrogel, stability, transmission electron microscopy, methacrylation, rheology, gelatin methacrylate

## Abstract

As the most abundant protein in the extracellular matrix, collagen has become widely studied in the fields of tissue engineering and regenerative medicine. Of the various collagen types, collagen type I is the most commonly utilised in laboratory studies. In tissues, collagen type I forms into fibrils that provide an extended fibrillar network. In tissue engineering and regenerative medicine, little emphasis has been placed on the nature of the network that is formed. Various factors could affect the network structure, including the method used to extract collagen from native tissue, since this may remove the telopeptides, and the nature and extent of any chemical modifications and crosslinking moieties. The structure of any fibril network affects cellular proliferation and differentiation, as well as the overall modulus of hydrogels. In this study, the network-forming properties of two distinct forms of collagen (telo- and atelo-collagen) and their methacrylated derivatives were compared. The presence of the telopeptides facilitated fibril formation in the unmodified samples, but this benefit was substantially reduced by subsequent methacrylation, leading to a loss in the native self-assembly potential. Furthermore, the impact of the methacrylation of the collagen, which enables rapid crosslinking and makes it suitable for use in 3D printing, was investigated. The crosslinking of the methacrylated samples (both telo- and atelo-) was seen to improve the fibril-like network compared to the non-crosslinked samples. This contrasted with the samples of methacrylated gelatin, which showed little, if any, fibrillar or ordered network structure, regardless of whether they were crosslinked.

## 1. Introduction

Collagen, both in its native form and also in its denatured form, gelatin, is able to form highly hydrated networks, termed hydrogels, which have been widely used in regenerative medicine, through printing or casting [1,2], to create 3D environments for cell proliferation and differentiation [3,4]. Collagen is suitable for use in regenerative medicine because it is abundant in the extracellular matrix (ECM), where it self-assembles and interacts with other collagens and with a wide range of other ECM molecules and cells [5,6].

The collagen family consists of 28 distinct types [7], of which type I collagen, the main component of skin, tendon, ligaments, and bone, is the most abundant. Many collagens are only minor components of the ECM, albeit with specific roles. All collagens share a common structural motif, the triple-helix, in which three individual chains, each in a left-handed polyproline II-like helix, are wound together to form a right-handed, rope-like triple helix [8]. This gives type I collagen a fairly rigid, rod-like structure, approximately 280 nm in length [8]. The amino acid sequence controls the secondary and tertiary molecular structure, with collagens that are characterised by a repeating (Gly-Xaa-Yaa)n sequence; only Gly is small enough to fit within the triple-helical conformation [8]. The Xaa and Yaa positions are most commonly proline and hydroxyproline (Hyp). Furthermore, structural constraints mean that the side chains of the residues in the Xaa position are more accessible, while those in the Yaa are less accessible due to steric hindrance. In denatured collagen, gelatin, the individual chains are flexible and have fewer steric constraints.

A major property of type I collagen is that when heated to 37 °C at neutral pH, purified, soluble type I collagen molecules contain sufficient binding domains to reform into fibrils, which, in turn, entangle to form a hydrogel [9]. During the biosynthesis of collagen, when molecules transit from the cell to the ECM, N- and C-terminal propeptide domains are removed, leaving short, 14- and 26-residue non-triple-helical telopeptide domains at each end of the collagen molecule. The 26-residue C-telopeptide is important in the initiation and rate of fibril formation [10] and, hence, in gelation. Type I collagen lacking the telopeptide domains, due, for example, to enzyme extraction, forms gels much more slowly [11]. Another factor that further limits fibril formation is whether the type I collagen is modified via, for example, succinylation, methylation or deamidation [12].

In order to create 3D collagen templates (scaffolds) that are mechanically stable, stabilisation by chemical or physical crosslinking methods is required. While many stabilisation methods are more suited to the treatment of dry collagen sponges or membranes [5], few are readily applicable to in situ crosslinking, as is required for 3D printing structures, in which rapid gelation is required [13]. One method that has gained importance is the prior methacrylation of gelatin [14] or collagen [15], followed by exposure to UV light with an initiator present, which allows casting or printing. The initial and preferred method of methacrylation for collagen was achieved using methacrylic anhydride [15], although other approaches, involving reactions using carbodiimides [16] and N-hydroxysuccinimide derivatives, are possible [17].

The presence of methacrylate groups, however, has the potential to change or interfere with the native fibril-forming process. This paper, therefore, examines the effects of modifying both telo- and atelo-collagens on the resulting fibril and subsequent network structures in photoinitiated gels, and compares these to the structures found in un-modified, collagen-based hydrogels and with those formed in methacrylated gelatin.

## 2. Materials and Methods

### 2.1. Collagen Samples

Rat tail telo-collagen was obtained from tail tendons cut into small sections (~10 mm) and suspended at ~5 mg/mL in 50 mM acetic acid (Chem Supply, Port Adelaide, South Australia, Australia) adjusted to pH 2.5 with HCl (Sigma Aldrich, St. Louis, MO, USA), at 4 °C for 16 h [18]. For preparation of atelo-collagen, pepsin was added at 1 mg/mL. After digestion, samples were clarified by centrifugation at 7500× *g* for 120 min at 4 °C. Collagen was precipitated by addition of NaCl (Sigma Aldrich, St. Louis, MO, USA) solution to 0.7 M, and left standing at 4 °C for 16 h. The precipitate was collected by centrifugation, 2500× *g* for 30 min, then dialysed exhaustively against 20 mM acetic acid and freeze-dried.

Deamidated collagen was prepared by incubating purified telo-collagen with 5% *w*/*v* NaOH (Sigma Aldrich, St. Louis, MO, USA) in saturated Na_2_SO_4_ (Sigma Aldrich, St. Louis, MO, USA) for 4-days [12] and was recovered by neutralising, followed by dialysis and freeze-drying, as previously described [19].

Methacrylated collagen was prepared using telo-and atelo-collagen samples dissolved at 3 mg/mL in 20 mM acetic acid and then adjusted to 200 mM NaCl and taken to pH 7.5 with NaOH, all at 4 °C. A 5-fold molar excess of methacrylic anhydride was added and the pH was maintained at 7.5 using NaOH for 4 h. The reaction was left overnight for completion, and then samples were dialysed exhaustively against 20 mM acetic and freeze-dried. The extent of collagen modification was determined in triplicate using a 2,4,6-trinitrobenzene sulfonic acid (TNBSA) assay (Thermo Scientific, Waltham, MA, USA) [20], following the manufacturer’s instructions.

Methacrylated gelatin (GelMA) was also prepared using a 5-fold molar excess of methacrylic anhydride in 200 mM NaCl at pH 7.5, but at 22 °C, using 300 g Bloom porcine skin gelatin (Sigma Aldrich, St. Louis, MO, USA).

### 2.2. Circular Dichroism

Collagen and GelMA samples were dissolved in phosphate buffered saline (PBS) at 0.1 mg/mL at 5 °C. Circular dichroism (CD) spectra were collected using samples in 1-mm cuvettes using a JASCO J-815 instrument (Jasco, Easton, MD, USA). Spectra were collected at 400–200-nanometer wavelengths, while temperature scans were performed at 222 nm at 0.3 °C/min from 10–50 °C.

### 2.3. Rheology

Collagen samples were prepared by dissolving at 30 mg/mL in 20 mM acetic acid at 4 °C. GelMA samples were prepared by dissolving at 10 mg/mL in PBS. Rheology measurements were performed with a Physica MCR 301 Rheometer (Anton Paar, Graz, Austria) fitted with a Peltier temperature controller and connected to an EXFO Acticure 4000 light source. Measurements were obtained using a 15 mm 1° conical plate. Immediately prior to measuring, collagen samples were neutralised to pH of 7.25 ± 0.25 with 1 M NaOH, and the salt concentration was balanced to 1 × PBS using 10 × PBS solution. Samples of 0.1 mL were used and added to the Peltier stage at 4 °C, and set at a gap distance of 500 µm. Samples were then held for a pre-shear period to remove prior memory, using a shear rate of 5 s^−1^ for 2 min at 4 °C. Samples were then taken to 37 °C over 60 s and rheological shear observations were performed for 30 min at a fixed oscillatory frequency of 1 Hz and an amplitude of 1%. Control samples were also held at 4 °C for 30 min to establish that no unwanted rheological changes were occurring. In collagen samples that had been methacrylated, and GelMA samples, lithium phenyl-2,4,6-trimethylbenzoylphosphinate (LAP) (Sigma Aldrich, St. Louis, MO, USA) was dissolved at 0.05%. Samples were then added to the rheometer, and the pre-shear procedure performed, as described above. The samples were exposed to 400 nm light at 20 mW/cm^2^ for 60 s, immediately prior to commencing the 30-min rheological testing, again monitoring at a fixed frequency of 1 Hz and amplitude of 1%, at 37 °C.

### 2.4. Transmission Electron Microscopy

Unmodified collagen samples were gelled after neutralising and heating to 37 °C, as with the rheology measurements, however, the TEM samples were diluted (1 mg/mL) so that a weaker gel that could be pipetted onto an imaging grid. Methacrylated samples had LAP added at 0.05%, followed by exposure to 400 nm light at 20 mW/cm^2^ for 30 s prior to taking to 37 °C, following the approach used for the rheology measurements. Gelled samples were examined by transmission electron microscopy (TEM) using a Tecnai 12 transmission electron microscope (FEI, Eindhoven, The Netherlands) at an operating voltage of 120 kV. Samples were prepared on carbon-coated grids (EMSCF200H-CU-TH, ProSciTech, Kirwan, Australia) that were glow-discharged to render them hydrophilic. A 1-microliter drop of sample was applied to an upturned grid held in anti-capillary forceps over moist filter paper, and left for 1 min over moistened filter paper. Excess liquid was then removed with filter paper and the grid was then inverted onto a drop of 2% PTA stain, pH 6.9 on parafilm, for 1 min. The grid was removed from the stain, and excess stain was wicked away with filter paper. Grids were allowed to dry before viewing under the microscope. Images were recorded using a FEI Eagle 4k × 4k CCD camera with AnalySIS v3.2 camera-control software (Olympus, Tokyo, Japan).

## 3. Results

Samples of both telo- and atelo- rat collagen were readily prepared and methacrylated. The proportion of methacrylation was similar in both the telo- and atelo- collagen samples with 55% (±7%, n = 3) and 52% (±4%, n = 3), respectively, of the Lys residues present in the methacrylated collagens. These values contrasted with the 77% (±3%, n = 3) obtained for the GelMA, which was also modified by the same excess of methacrylic anhydride. These results were consistent with other reports, in which a similar molar excess of reagent was used [21,22].

The chemical modification of collagens can lower the stability of the triple helix [12,23]. This was examined in the present samples through CD spectroscopy (Figure 1), which found that both the methacrylated samples gave similar spectra to the unmodified collagens. The wavelength of the maximum absorption was observed at 222 nm for both.

The thermal response curves, measured at 222 nm (Figure 2), indicated that the methacrylation of both the telo-collagen (Figure 2A) and the atelo-collagen (Figure 2B) led to the destabilisation of the triple helix, with around 2 °C decreases in thermal stability in both cases. The atelo-collagen samples were found to have marginally higher thermal stability compared to those of the telo-collagen, a phenomenon that has been observed previously [24].

Turbidity has frequently been used to study the formation of collagen gels by heating cold, neutral solutions to 37 °C [11,25]. Studies have demonstrated a lag phase (where no change in modulus occurred as molecules heated and began or organise), of around 10 min, depending on the conditions, which suggests that fibril nucleation occurred, followed by an increase in turbidity as fibril formation, thickening and growth continued [26,27]. In the present study, however, photo-rheology was used as it enables quantifiable changes in viscosity and modulus following the irradiation of methacrylated collagen samples for observation. This approach was validated using samples of telo-, atelo- and deamidated telo- rat tail type I collagen (Figure 3A). These rheological data showed that the telo-collagen exhibited rapid gel formation, after a lag period of about 11 min, following the increase in the temperature from 4 to 37 °C, which was consistent with the turbidity data. By contrast, the atelo-collagen showed slower gelation after an extended lag period. The data for the deamidated collagen showed that little, if any, fibril formation or association occurred (Figure 3A). Thus, the absence of the telopeptides significantly decreased both the rate of fibril formation and the modulus that was obtained after 45 min by almost 50%.

The methacrylation of both collagen samples resulted in a reduced rate of viscosity increase during the thermal gelation process. This was most notable for the methacrylated telo-collagen (Figure 3B), where the reduction in the change in viscosity was about 80%, falling to a level that was much lower than that of the unmodified atelo-collagen. However, the extent of the reduction upon ~50% methacrylation modification was less than for the deamidated collagen where limited, if any, fibril formation occurred. There was also a reduced change in viscosity for the methacrylated atelo-collagen of about 80%, giving a particularly low value as the original unmodified material showed a limited viscosity increase over the 45 min of heating (Figure 3C). In all cases, except for the deamidated collagen, visual inspection found that some form of gel formed, although these forms were typically weak, except for the unmodified telo-collagen, which formed a firm gel. The change in modulus after UV irradiation of the methacrylated collagens was much more rapid (Figure 4), reaching comparable modulus levels around four times more rapidly than any of the thermally gelled materials. A summary of the rheological data indicates that statistically significant differences changes were found when different materials were examined (Figure 5).

The structures of the collagen fibrils, aggregates, and networks in the various hydrogels, with and without UV crosslinking, were examined by TEM (Figure 6). These data showed that the fibrils in the hydrogels formed by thermal treatment of telo-collagen solution were large in diameter, and showed the well-ordered, asymmetrically banded, staggered overlap structure that is observed in native tissues [28,29] (Figure 6A). The asymmetry of these fibrils, a head–tail to head–tail arrangement, was evident from the minor banding seen within the major, 67 nm banding pattern. These large structures overlapped and entangled to provide a network with the stability required for gel formation (Figure 6B). The diameters of the individual fibrils were typically between 20 and 50 nm, similar to the size seen in immature tissue [30], but not always as well formed. Frequently, larger structures that comprised many small fibrils that had coalesced and entangled to give larger assemblies were seen (Figure 6A,B). By comparison, the fibril-like structures formed by the atelo-collagen were smaller in diameter, and more numerous (Figure 6C), and did not typically show a normal staggered overlap structure (Figure 6C). These fibril-like structures (Figure 6C) could potentially include symmetric, head–tail to tail–head structures, which are found in fibrous-long-spacing (FLS) packing [28]. These would have shown a different staining pattern, but this was also not seen. Rather, the larger fibre bundles of the atelocollagen formed an extended network that was sufficient to stabilise a weak gel, and were augmented by a range of thin structures (Figure 6C), which were possibly only a few molecules thick, which may have been insufficient to show any banded stain pattern [29].

The collagen networks of methacrylate-modified collagens after thermal but not UV gelation were distinct from those of unmodified collagens (Figure 6D,E). Long, extensive fibrillar structures were not observed in either the telo- or the atelo-collagen (Figure 6D,E). Some laterally aggregated structures were present, some of which showed apparent non-specific interactions that may have contributed to the increase in viscosity and could have contributed towards a weak gel network. The images obtained revealed no collagen structure, suggesting that most of the collagen was present in the gel as single molecules that were not resolved by the TEM.

The hydrogel structures formed by the methacrylate-modified collagens after UV gelation were distinct from those of the thermally treated samples (Figure 6F,G). The methacrylated telo-collagen showed a mixture of large, poorly formed fibrous bundles, giving a coarse network, which was associated with many small, thin fibrous entities. Small, round structures were also observed, which were interpreted as being due to the LAP catalyst as they were only present in the samples that had LAP added (Figure 6F,G,I) and have been observed previously in resilin samples in which LAP was present [31]. The methacrylated atelo-collagen network (Figure 6G) was similar to that observed with the methacrylated telo-collagen, except that the fibrous bundles were the major component, and the fine network of smaller fibres was not as extensive.

The hydrogel formed by methacrylated gelatin in the absence of UV irradiation (Figure 6H) showed a different, distinct structure. There was no evidence of any re-formed structures of the same length as a collagen molecule. Rather, a range of smaller, indistinct structures of various sizes was present as part of a semi-continuous assembly. The hydrogel formed by the methacrylated gelatin after UV irradiation (Figure 6I) also showed few, if any, apparent structures. Zones of denser, amorphous material were present, along with the structures interpreted as the LAP addition.

## 4. Discussion

Hydrogels have gained significant interest in tissue engineering and regenerative medicine for the delivery of cells to form new replacement tissues [32,33]. The structure and chemistry of hydrogels can affect the performances of cells [34,35,36]. For example, the stiffness of hydrogel networks can lead to variations in cellular responses. Various biofabrication and regenerative medicine approaches rely on hydrogels with specific Young’s moduli to best facilitate cellular differentiation, diffusion and mechanical robustness [34,35]. The moduli of native soft tissues, such as those of the brain (0.1–1 kPa), muscle (~10 kPa) and pre-mineralised bone (30 kPa), have been established; recreating these environments is essential to directing stem-cell lineage. Stem cells can differentiate towards these lineages via cues such as scaffold stiffness [37,38].

For collagens and gelatines, the structure, organisation, density, and extent of additional crosslinking can all affect and provide control over the Young’s modulus to allow the creation of a template with tunable stiffness. The use of these biopolymers in the formation of hydrogel templates can provide the further advantage of providing suitable cellular interactions [5]. Type I collagen contains numerous different cellular interaction sites [39,40]. Some of these binding sites occur on isolated molecules, while others require a larger, properly formed fibrillar site to be present [41]. However, if the collagen is chemically modified, such as through the addition of methacryl functionality, the extent of some interactions, especially those dependent on fibrillar structures, could be reduced or lost [42]. In gelatin, the denaturation of collagen leads to the loss of conformation-dependent sites. However, linear RGD sequences become available, after being hidden by the triple-helical conformation that internalises the Gly residue, so that integrin binding can occur, albeit by a different path to that seen on triple-helical collagen [43].

The formation of collagen hydrogels through warming neutral solutions is well established [9,44], with the variations in the hydrogel structure determined by the source [45], concentration [46], incubation temperature [47], pH [48] and ionic strength [49] of the collagen solution. For laboratory experiments, rat-tail-tendon type I collagen is a preferred source, as this collagen can be acid-extracted so that it retains telopeptides, which enhance fibril formation [11] (Figure 3A). Other collagens that could be acid-extracted in good yield with intact telopeptides include marine collagens [50] and chicken collagen [51]. The presence of telopeptides from xenogeneic species, however, may not be preferable for clinical applications as telopeptides can cause immune responses [52].

The telo-collagen in the thermally produced hydrogels reformed into fibrils analogous to those in native tendons and other tissues, showing an asymmetric banded structure on the TEM (Figure 6A,B). Each fibril was composed of triple-helical collagen molecules arranged in a staggered, overlapped array, resulting in a characteristic 67 nm banding periodicity in which all the molecules were oriented in one direction. Each repeating band consisted of a 30 nm “overlap” zone and a 37 nm “gap” zone, which led to the observed banding through different rates of stain uptake [53]. The collagen in these gels has a native, strong interaction with cells, which leads to the consolidation of the hydrogel into a more tissue-like structure [54].

Atelo-collagen, which is generally obtained through the digestion of tissues by pepsin [55], can also form into hydrogels, but the rate of formation is much slower (Figure 3A). The TEM showed that the structure was less well organised (Figure 6C) but still had a fibrous network that was able to form a hydrogel that could be reorganised and consolidated [56]. However, hydrogels from either telo- or atelo collagen that is subsequently cross-linked retain their size when incubated with cells and do not readily consolidate [57]. The fibrous structure observed in these atelocollagen hydrogels (Figure 6C) was poorly organised and lacked the ordered, asymmetric, staggered overlap structure of native fibrils. A mixture of other centrosymmetric structures may have been present, with at least four fibrous-long-spacing (FLS) packing modes having been observed [58]. In addition, smaller collagen structures were present, also forming a quasi-network structure (Figure 6C). In some cases, the visible material was comparable to the length of a single collagen molecule, but probably comprised in-phase aggregates of collagen molecules, analogous to the segment-long-spacing (SLS) assemblies that can be observed under specific conditions [59]. In other cases, end-to-end assemblies of these structures occurred, leading to smaller fibril-like structures (Figure 6C).

Typically, modified collagens, such as deamidated collagen, do not form into fibrils (Figure 3A), either because of changes to the charge distribution or because of steric hinderance, and probably remain as single molecules. In the present study, neither the telo- nor the atelo-methacrylated collagens showed a major propensity for thermal hydrogel formation, although there was a slight degree of gelation, unlike in the fully deamidated collagen. This reduction was not due to denaturation, as the modified collagens were still triple-helical (Figure 1 and Figure 2), albeit with slightly reduced thermal stability.

Individually, both these modified collagens, although modified using an excess of reagent, were just over 50% modified based on the number of available Lys residues. Similarly, the prepared GelMA was modified to around 80%. The reactivity of Lys residues depends on their pKa values, with those with lower pKa values reacting more readily [60]. In any given protein, the pKa values of the Lys residues vary with the surrounding tertiary structural environment, although often with only small differences [61]. In collagens, Lys residues are found preferentially in the Yaa position [62] and of these, several may be subject to secondary modification to give hydroxylysine, some of which may be additionally glycosylated, potentially further reducing their reactivity. The lower degree of reaction of the collagen compared to the GelMA reflects the changes in the tertiary structure such that the Lys residues of the gelatin were more readily available, while the reactivity of some, potentially those with secondary modifications, was still low, leading to incomplete modification.

The ultrastructure of the methacrylated collagen hydrogels formed by the thermal treatment, as shown by the TEM (Figure 6D,E), were quite distinct from those of the unmodified collagens. Both preparations showed structures that were the length of individual collagen molecules, but were probably wider than single molecules. This suggests that lateral, in-phase assemblies were formed, analogous to SLS aggregates, but not of sufficient width or diameter to enable any fine structure banding to be visible. Additional interactions, however, could lead to a network sufficiently fine to stabilise hydrogels.

By contrast, the networks formed by the UV crosslinking of the methacrylated telo- and atelo-collagens were distinct (Figure 6F,G). The use of UV prior to the thermal treatment allowed little time for any natural associations to occur and the crosslinking led to extensive fibrous material formation that lacked any apparent order. This fibrous material further aggregated to give larger fibrous ropes. For the methacrylated telo-collagen, a significant amount of fine fibre network was also present, but this was less significant for the methacrylated atelo-collagen material (Figure 6F,G).

The TEM also showed that the structure of the methacrylated gelatin hydrogel in the absence of UV irradiation (Figure 6H) was quite distinct from the collagen-based hydrogels. No structures equivalent to full-length collagen molecules were seen, but an array of smaller structures was present. When a normal, unmodified gelatin hydrogel forms, small sections of the individual, denatured chains reassemble to form triple-helical segments, although it is unlikely that these relate to the natural chain organisation [63]. These triple-helical segments allow an initial fine network to form, providing the junctional domains that enable gel formation. With increasing time, the triple-helical domains coalesce to form larger structures, providing what is termed a coarse network [63]. In this non-crosslinked structure, the association of randomly organised chains occurs, while the re-formation of triple-helical segments consolidates the structure. In the samples that were UV-crosslinked, there was no apparent structure associated with the gel, and only regions of amorphous density were seen. However, when a gel is formed, some form of network must be present, with chains possibly only of single molecules. Nevertheless, the high degree of methacrylation, 80%, would have led to the formation of a large number of crosslinks. Gelatin solution is readily digested by proteases, leading to small, non-gelling fragments. When crosslinked methacrylated gelatin is incubated with cells, which typically have a range of cell surface proteases [64], as well as potentially secreting proteases, the extent of crosslinking limits the solubilisation of the gel, either directly, or by limiting the rate of cellular migration until sufficient remodeling has occurred [21].

The methacrylation of a denatured collagen, gelatin, has proven to be a popular method for tissue engineering and for the production of bioinks, where gelation induced by photochemical methods allows the rapid rates of crosslinking and stabilisation that are needed for successful bioprinting. The mechanical properties of methacrylated gelatin and, consequently, its cell and tissue interaction properties, can be tuned by adjusting the nature of the material though variations in the concentration and the extent of crosslinking [21]. Cells bind to gelatin networks, possibly due to the presence of otherwise cryptic RGD sites that are revealed during the denaturation of the collagen. Native collagen has a broader range of highly specific binding sites, including for integrin binding, through a conformationally dependent sequence, while being considerably more stable towards general proteolysis. Native collagen was therefore examined in this study as an alternative material for forming a biologically-based hydrogel network. As with methacrylated gelatin, it was expected that the concentration and extent of crosslinking would influence the characteristics of the hydrogel. For collagen, however, the nature of the networks that are formed can be variable, depending on the conditions under which the collagen extraction is performed. Thus, the hydrogels formed from acid-extracted telo-collagen can be distinct from those of enzyme-extracted atelo-collagen, from which the telopeptides are removed during enzyme solubilisation, giving a further variable for hydrogel formation that is not present in gelatin-based hydrogels.

In conclusion, this study highlights the differences in ultrastructure that were produced in collagen hydrogels that were extracted by varying techniques and as a result of methacrylation. The presence of telopeptides was seen to facilitate fibril formation to a large degree, compared to atelo- forms of collagen. Once methacrylated, the triple helix was seen to be destabilized and to undergo denaturation at lower temperatures compared to the unmodified forms. Methacrylation was also seen to impact fibril formation in both the methacrylated telo- and methacrylated atleo- conditions. Future studies could be targeted to understand how various collagen ultrastructures impact cellular function and the mechanical properties of these hydrogel scaffolds.

## Figures and Tables

**Figure 1 polymers-14-01775-f001:**
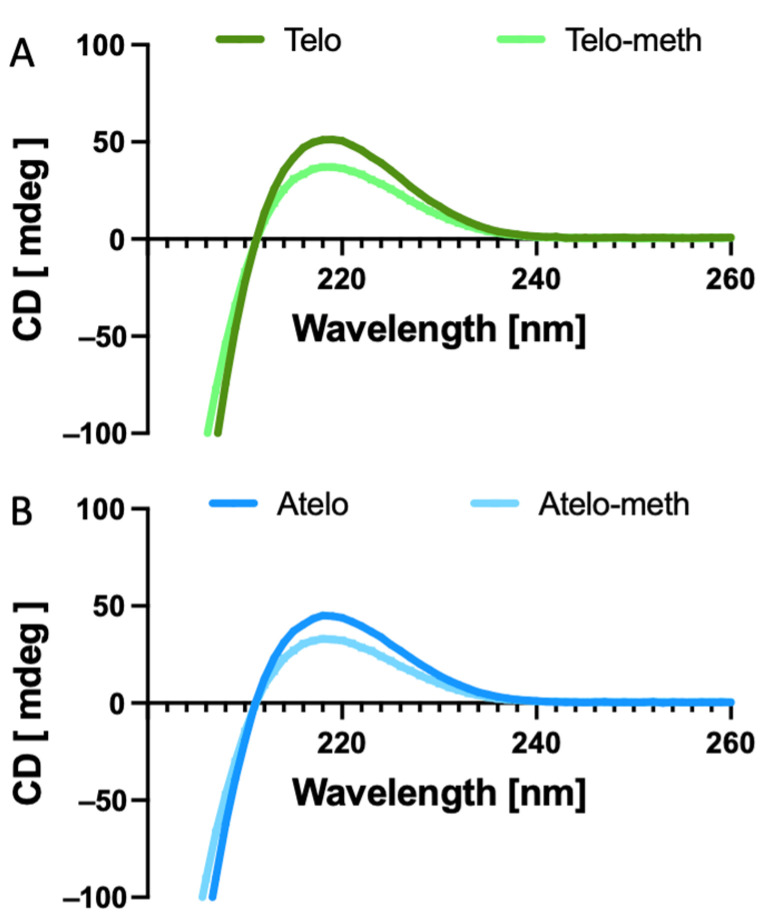
CD spectra of collagens and modified collagens. (**A**) Telo-collagen and methacrylated telo-collagen. (**B**) Atelo-collagen and methacrylated atelo-collagen. In both cases, the modified collagens showed the lower ellipticity.

**Figure 2 polymers-14-01775-f002:**
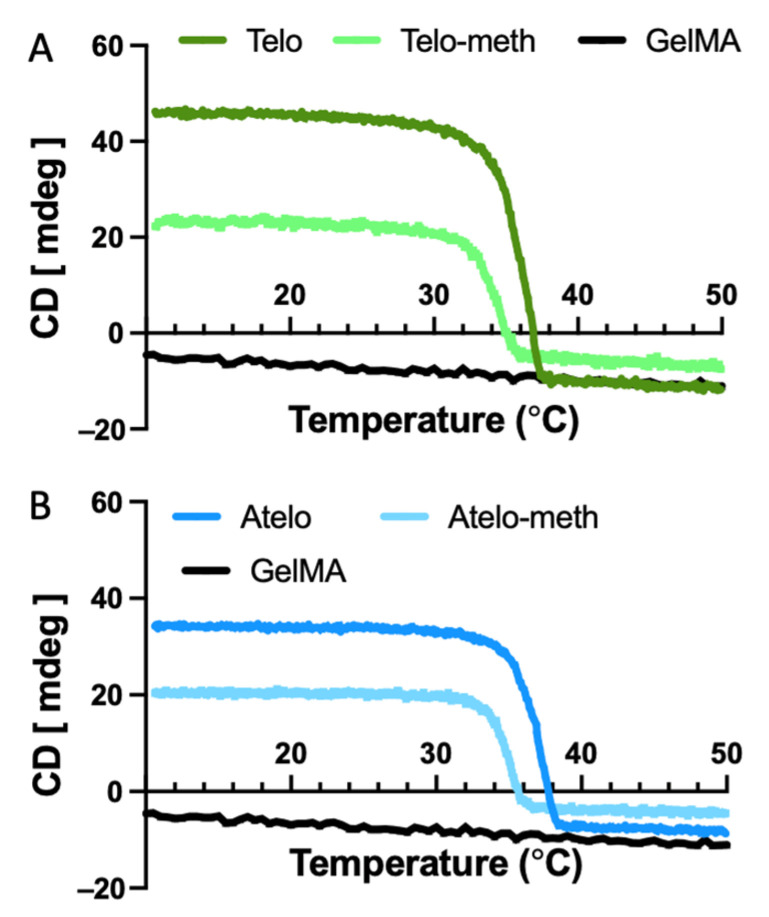
Thermal denaturation of collagens and modified collagens measured using CD spectroscopy at 222 nm. (**A**) Telo-collagen and methacrylated telo-collagen. (**B**) Atelo-collagen and methacrylated atelo-collagen. In both cases, the modified collagens showed the lower stability.

**Figure 3 polymers-14-01775-f003:**
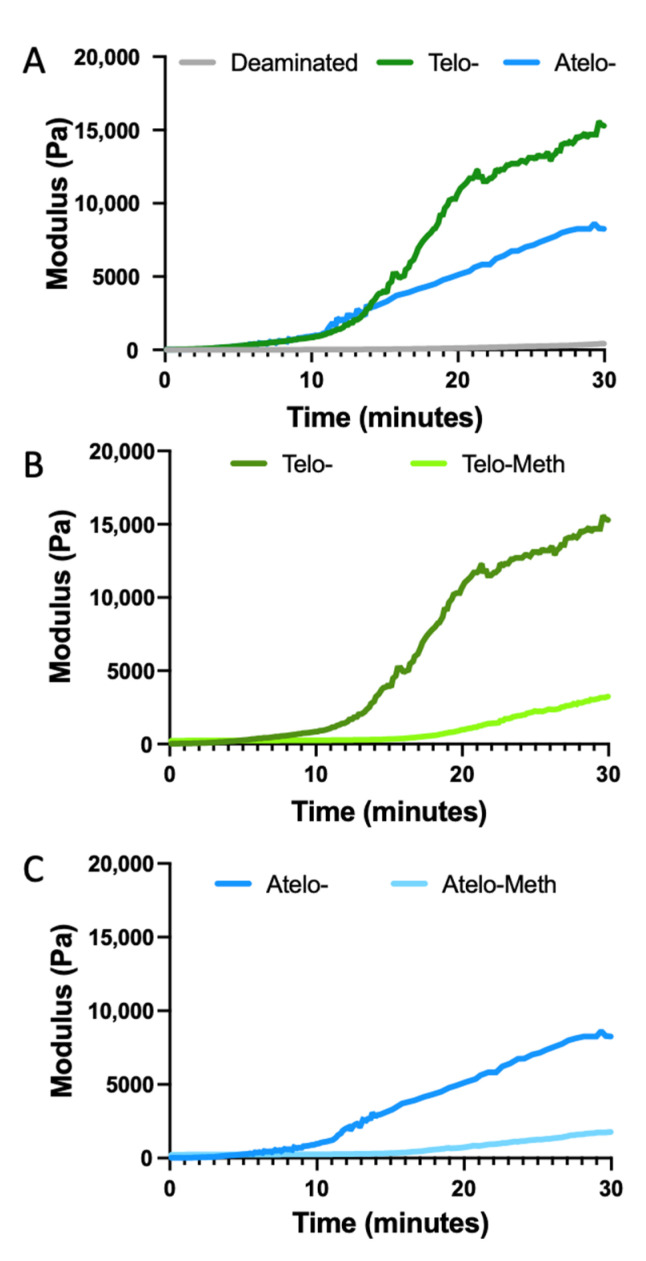
Rheological examination of collagen and modified collagens. (**A**) Telo-collagen, atelo-collagen and deamidated telo-collagen. (**B**) Telo-collagen and methacrylated telo-collagen without UV irradiation. (**C**) Atelo-collagen and methacrylated atelo-collagen without UV irradiation.

**Figure 4 polymers-14-01775-f004:**
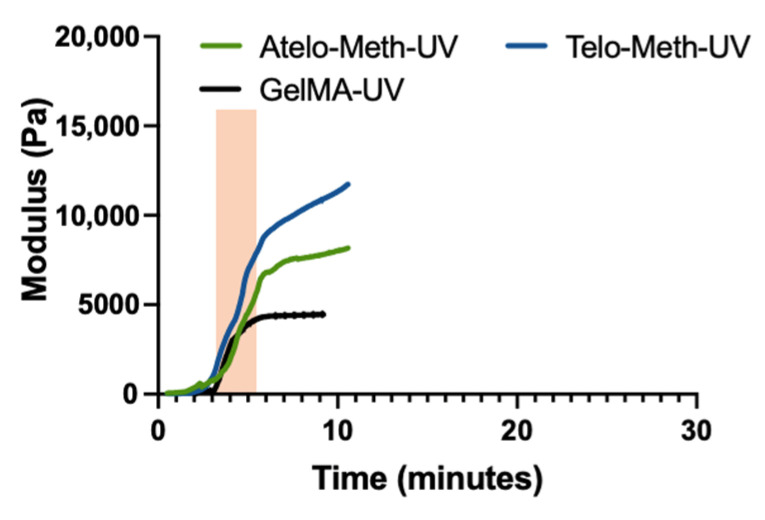
Rheological examination of photo-crosslinking kinetics of methacrylated telo-collagen and methacrylated atelo-collagen, following exposure to UV irradiation. The orange bar represents the UV exposure period.

**Figure 5 polymers-14-01775-f005:**
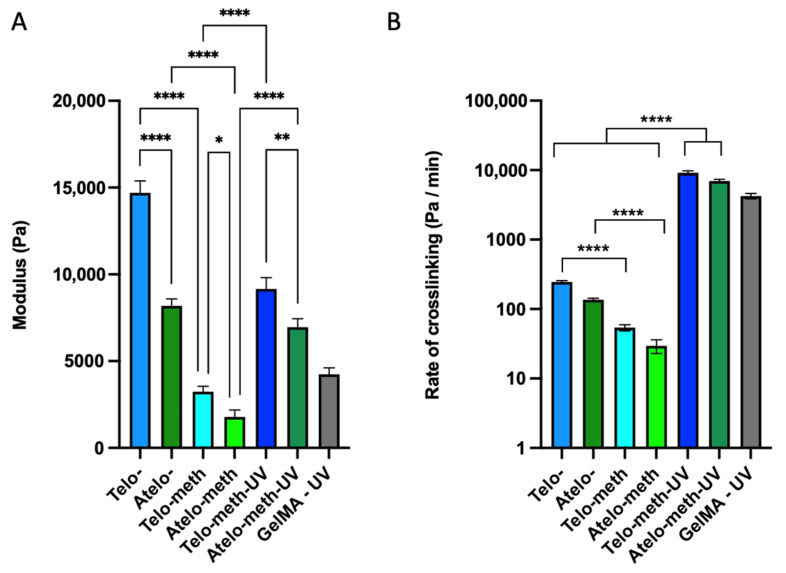
Summary of the rheological crosslinking scans. (**A**) The modulus immediately following the crosslinking period; for fibrillogenesis samples this occurred at 50 min, and for the UV samples this occurred at 5 min (following the 1-min UV exposure). (**B**) The rate of crosslinking during the active phase (50-min window for the fibrillogenesis, and 1-min window for the photo-crosslinked samples). Results of the one way ANOVA show significance between groups with * = *p* < 0.05, ** = *p* < 0.01 and **** = *p* < 0.0001.

**Figure 6 polymers-14-01775-f006:**
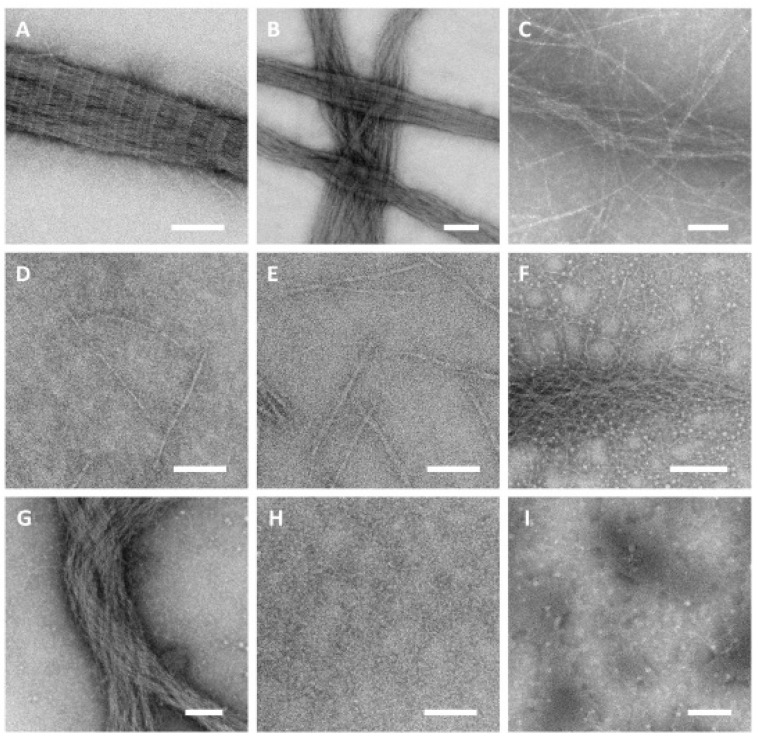
TEM of collagens, methacrylated collagens and GelMA samples after thermal and UV-induced gelation. (**A**) Telo-collagen. (**B**) Telo-collagen. (**C**) Atelo-collagen. (**D**) Methacrylated telo-collagen without UV irradiation. (**E**) Methacrylated atelo-collagen without UV irradiation. (**F**) Methacrylated telo-collagen with UV irradiation. (**G**) Methacrylated atelo-collagen with UV irradiation. (**H**) GelMA without UV irradiation. (**I**) GelMA with UV irradiation. Bars = 100 nm, excepting (**B**,**F**), where Bar = 200 nm.

## Data Availability

Not applicable.

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
