# Peer review of "Variation in Hydrogel Formation and Network Structure for Telo-, Atelo- and Methacrylated Collagens"

_polymers, 2022, doi:10.3390/polym14091775_

Round 1

Reviewer 1 Report

The paper examines the effects of 78 modifying both telo- and atelo-collagens on the resulting fibril and subsequent network 79 structures in photoinitiated gels and compares these to the structures found in un-modi-80 fied collagen-based hydrogels and with those formed in methacrylated gelatin.

Authors provided results through rheology, circular dicroism and TEM. Although very few techniques were carried out, the experiments were well explained and discussed properly. The discussion regarding rheology and its correlation with collagen structure is coherent and the choice of the experimental conditions is adequate for the purpose of investigation of collagen structure. Some results were quite obvious considering the chemical characteristics of gelMA and collagen. 

Reviewer 2 Report

The topic of the article is interesting, and the authors conducted good quality work; the writing is clear and the study method appropriate. I consider this paper of conspicuous interest to readers.

The title is correct and clear, and the summary is well-ordered and compressive. I do not have any consideration in this section.

However it does have shortcomings that need to be addressed before any further considerations should be made.

First, the use of English needs to be corrected. There are far too many grammatical and style errors throughout the text. 

In general, the results and the discussion are well expressed. 

Conclusion section is missing!
This section needs to be adedd. However, there will be no future studies? I recommend the authors put de something adds two sentences referring them to future studies.

I recommend that authors follow the same format for all references.

For improvement, the manuscript should be revised according to the above suggestions and those of other reviewers. In my honest opinion, I suggest a minor revision of the article. The authors have done work that provides interesting results.

Round 2

Reviewer 2 Report

The authors improved the manuscript according to reviewers' commenst and suggestions.